# Acute Pain Management Pearls: A Focused Review for the Hospital Clinician

**DOI:** 10.3390/healthcare11010034

**Published:** 2022-12-22

**Authors:** Sara J. Hyland, Andrea M. Wetshtein, Samantha J. Grable, Michelle P. Jackson

**Affiliations:** 1Department of Pharmacy, OhioHealth Grant Medical Center, Columbus, OH 43215, USA; 2Department of Pharmacy, Cleveland Clinic Fairview Hospital, Cleveland, OH 44111, USA; 3Hospice and Palliative Medicine, OhioHealth Grant Medical Center, Columbus, OH 43215, USA

**Keywords:** pain management, opioid stewardship, postoperative pain, multimodal analgesia, transitions of care, opioid-related adverse effects, acute pain, opioid use disorder, enhanced recovery, opioids

## Abstract

Acute pain management is a challenging area encountered by inpatient clinicians every day. While patient care is increasingly complex and costly in this realm, the availability of applicable specialists is waning. This narrative review seeks to support diverse hospital-based healthcare providers in refining and updating their acute pain management knowledge base through clinical pearls and point-of-care resources. Practical guidance is provided for the design and adjustment of inpatient multimodal analgesic regimens, including conventional and burgeoning non-opioid and opioid therapies. The importance of customized care plans for patients with preexisting opioid tolerance, chronic pain, or opioid use disorder is emphasized, and current recommendations for inpatient management of associated chronic therapies are discussed. References to best available guidelines and literature are offered for further exploration. Improved clinician attention and more developed skill sets related to acute pain management could significantly benefit hospitalized patient outcomes and healthcare resource utilization.

## 1. Introduction

Acute pain plagues the majority of hospitalized patients at some point during their clinical course [1]. Despite its commonness to inpatient care, acute pain management has been underrepresented in medical didactic curricula and experiential training, leaving many healthcare providers ill-equipped to effectively manage pain [2]. Meanwhile, the consequences of both uncontrolled pain and of indiscriminate opioid prescribing increasingly torment patients, healthcare systems, and communities [3,4,5,6,7,8,9]. The populations of hospitalized patients with preexisting opioid tolerance, chronic pain, or opioid use disorder (OUD) are also increasing alongside limited availability of applicable specialists, further challenging prescribers and straining healthcare resources [4,10].

We therefore underscore acute pain management as a necessary function of inpatient providers and seek to support diverse practitioners in updating or refining applicable knowledge and skills in this realm. Herein, the interprofessional collaborators have provided our “top ten” recommended clinical pearls for acute pain management, with each including discussion, key considerations, and visual summaries of recommendations. Our specific aim is to provide a focused narrative review and actionable point-of-care resource for acute pain management to inpatient providers. The ten clinical pearls discussed herein include:Uncontrolled pain worsens patient outcomes and healthcare costs—adopt a consistent, systematic, and holistic approach to acute pain management.Not every patient is an ideal candidate for every medication, but every patient in pain is a candidate for multimodal analgesia optimization.There are tremendous benefits to employing anti-inflammatories and few good reasons to withhold them in the management of acute pain.Gabapentinoids have a complex risk/benefit ratio and decision-making should be nuanced.Low-dose ketamine is a powerful analgesic even in opioid-tolerant patients and is generally well-tolerated with appropriate institutional protocols.Empiric opioid regimens should include consideration of optimal agent selection, dosing, route of administration, and supportive therapies.Pain regimens should be evaluated and adjusted at least daily through multidimensional pain assessments to optimize efficacy and safety endpoints.Patients with opioid tolerance, chronic pain, and/or opioid use disorder require higher opioid doses and more supportive therapies.When used for acute pain, patient-specific plans for opioid tapering and harm reduction should be developed and supported across the care continuum.Methadone and buprenorphine should almost always be continued throughout acute pain episodes, but naltrexone must be stopped.

The reader is referred elsewhere for comprehensive reviews and applied examples of foundational analgesia resources, including equianalgesic opioid calculations and types of pain (e.g., nociceptive, visceral, or neuropathic, all of which may play a role in acute pain referred to in this piece) [11,12].

## 2. Clinical Pearl #1: Uncontrolled Pain Worsens Patient Outcomes and Healthcare Costs—Adopt a Consistent, Systematic, and Holistic Approach to Acute Pain Management

While pain management may frequently be associated with patient perception surveys in the minds of hospital-based practitioners, its importance to patient outcomes extends far beyond satisfaction with care delivery. Uncontrolled acute pain triggers a complex neurohormonal cascade that is toxic to nearly every organ system, as evidenced by increased rates of renal and gastrointestinal dysfunction, infection, cardiopulmonary and thrombotic complications, impaired wound healing, adverse psychological effects, and poorer functional recovery and quality of life [3,4,13]. Further testament to this concept is in the benefits of analgesic interventions beyond improved pain control: As one example, regional anesthesia has been found to reduce many postoperative morbidities and mortality [14,15]. The clinical and socioeconomic impacts of uncontrolled acute pain are vast, driven by longer lengths of stay, more readmissions, and increased risk for the development of persistent post-discharge pain and opioid use [3,5,9]. Acute pain management in hospitalized patients should therefore be a clinician and institutional priority as a key driver of patient outcomes and institutional resource utilization.

While opioids have an important role in acute (and chronic) pain management, they carry many toxicity risks, including pruritus, nausea and vomiting, constipation, ileus, urinary retention, delirium, sedation and respiratory depression. Such opioid-related adverse events (ORAEs) confer longer lengths of stay and significant economic burden [6,16]. Opioid agonists also induce μ-receptor desensitization and tolerance with repeated exposure, contributing to opioid withdrawal syndrome and opioid use disorder (OUD) [17,18].

Indiscriminate opioid prescribing has been criticized as fueling the modern U.S. opioid epidemic, yet many hospitalized patients report inadequate pain relief, signaling both overreliance on opioids and suboptimal pain management strategies by healthcare providers [1,7,8,19,20,21]. Herein lies the impetus for Opioid Stewardship Programs (OSPs), as providers and institutions must appreciate both the risks of unnecessary opioid exposure and of undertreated pain [22].


**KEY CONSIDERATION: **
*Opioid avoidance in acute painful conditions is an ill-advised goal as it is neither patient-centered nor evidenced-based; rather, multimodal analgesia and opioid stewardship should integrate into a patient-specific, data-driven approach to inpatient acute pain management.*


We therefore recommend all hospital-based healthcare providers adopt a consistent, systematic, and holistic approach to acute pain management as a daily patient care function, as depicted in Figure 1 and described further in subsequent sections.

## 3. Clinical Pearl #2: Not Every Patient Is an Ideal Candidate for Every Medication, but Every Patient in Pain Is a Candidate for Multimodal Analgesia Optimization

The most effective strategy for improving pain control and decreasing adverse medication events in this setting is multimodal analgesia. Such “rational polypharmacy” maximizes benefit while decreasing risk by combining lower doses of multiple medications with complementary mechanisms of action [21,23,24,25,26,27]. Hence, all patients experiencing acute pain should be prescribed appropriate, scheduled (i.e., around-the-clock, *not* “as needed”), nonopioid analgesics of different classes in addition to any opioid therapies.

This recommendation should not be conflated with one of prescribing all available nonopioids to patients indiscriminately or in an overly protocolized manner, however. For example, acetaminophen should form the backbone of most acute pain regimens due to its unique mechanism and wide safety margin, even in patients with chronic liver disease [28]. While it can be safely and effectively combined with other analgesics in a majority of patients [21,29], acetaminophen should be avoided in excessive amounts and in acute liver injury due to its known hepatotoxicity under these circumstances [30]. Anti-inflammatories, gabapentinoids, and NMDA antagonists will be discussed in subsequent sections.


**KEY CONSIDERATION: **
*Multimodal analgesia should be balanced and thoughtful, utilizing complementary mechanisms to improve pain control with less high-risk drug exposure, as opposed to overmedicating without regard to cumulative risks or patient-specific factors.*


Multimodal analgesia should utilize modest to moderate doses of however-many agents are sensible for the particular patient at that time, within the scope of available institutional protocols to ensure safe and meaningful use. Providers should consider pain features, risk factors for adverse events, drug interactions, and the global clinical status of the patient when initiating and adjusting multimodal analgesic regimens. At the institutional level, interprofessional stakeholders should proactively engage to create order sets or “pain management menus” that provide decision-support and cohesive operationalizing of a variety of multimodal modalities [12,22]. These may include:AcetaminophenAnti-inflammatoriesNeuropathic agents- gabapentinoids, serotonin reuptake inhibitors, anticonvulsantsCorticosteroidsNMDA antagonistsCentral alpha adrenergic agonistsSystemic anesthetics (e.g., intravenous lidocaine, inhaled anesthetics)Topical agents- lidocaine, anti-inflammatories, capsaicinRegional anesthetic modalities in concert with a specialist pain service (e.g., peripheral nerve blocks, neuraxial blocks)Physical therapy, cognitive/behavioral therapies, thermotherapies, and other nonpharmacologic modalities

There are multiple non-pharmacological therapies that can complement medication therapies. Three examples of complementary therapies with increasing evidence and uptake into inpatient practice include massage therapy, acupuncture, and music therapy. Massage therapy has been vigorously studied in regard to cancer pain, but the tenets can be applied to acute pain management. Therapeutic massage may increase blood and lymphatic circulation, decrease inflammation and edema, relax muscles, increase dopamine and serotonin levels, and reduce levels of anxiety, depression, anger and fear [31,32]. Acupuncture is a popular form of complementary medicine that assists to relieve symptoms and support bodily self-healing [31,33]. Lastly, music therapy is an emerging discipline that contributes to the elimination of psychological barriers and the restoration or improvement of physical and mental health. Music therapy has been found to decrease anxiety, which can lead to decreased use of analgesia [34,35,36,37,38]. These therapies have been used in acute pain management including across perioperative care, and should be offered concurrently with pharmacologic modalities to the extent possible within institutional capabilities [32,39].

## 4. Clinical Pearl #3: There Are Tremendous Benefits to Employing Anti-Inflammatories and Few Good Reasons to Withhold Them in the Management of Acute Pain

Non-steroidal anti-inflammatory drugs (NSAIDs), including selective COX-2 enzyme inhibitors like celecoxib, exert a potent analgesic effect in acute painful states that outperforms that of opioids when compared directly in randomized, double-blind trials [40]. This is likely due to their mechanism of action targeting a key source of pain for many acute painful processes, as opposed to only interfering with pain signaling [41,42,43,44]. Hospital providers should therefore recognize the significant benefits of NSAIDs to acute pain management and ensure patients receive them whenever appropriate.


**KEY CONSIDERATION: **
*NSAIDs target an important source of pain instead of just interfering with the perception of pain, making them among the most effective and important analgesics available in acute pain management.*


While many dogmatic safety concerns and perceptions exist with NSAIDs, most of these are no longer supported by published evidence and/or do not preclude the short-term use of NSAIDs to treat acute pain in hospitalized patients (Table 1). Importantly, the adverse event risks inherent to NSAIDs should be considered in the greater context of their benefits to improved analgesia, decreased opioid use, and enhanced recovery for the particular patient when making treatment decisions [45].

## 5. Clinical Pearl #4: Gabapentinoids Have a Complex Risk/Benefit Ratio and Decision-Making Should Be Nuanced

Gabapentinoids (gabapentin and pregabalin) are some of the most widely prescribed medications in the United States especially in the perioperative setting, where opioid sparing regimens as part of enhanced recovery after surgery (ERAS) protocols have become the standard of care [73]. The inclusion of gabapentinoids as part of an opioid sparing regimen is, in part, due to widely held beliefs that gabapentinoids are without drug–drug interactions and have large therapeutic indices [74]. There is also a common misperception that gabapentinoids are not as addictive as opioids and are therefore mentioned in many guidelines for the treatment of pain for a multitude of patient populations, although their use is often off-label [75].

When considering a gabapentinoid for your patient, it is important to understand their pharmacology and metabolism. Gabapentinoids exert their analgesic effect by binding to the alpha-2-delta subunit of N-type voltage-gated calcium channels in the central nervous system, decreasing the influx of calcium through the channels, and subsequently reducing the outflow of excitatory neurotransmitters to mitigate neuropathic pain [74,76]. The reduction in outflow of excitatory neurotransmitters is the premise behind gabapentinoids use in enhanced recovery protocols. Both gabapentin and pregabalin are renally eliminated unchanged in the urine and require dose adjustments in patients with altered kidney function. It is important to note that in patients who are undergoing hemodialysis, greater than 40% of the medication remains, posing safety risks in those patients who miss dialysis or who are unable to tolerate a full dialysis session as they will be at risk for accumulation of the medication [74].


**KEY CONSIDERATION: **
*Gabapentanioids are heavily renally eliminated and require dose adjustment in altered kidney function. Patients undergoing dialysis who are apt to miss appointments or who are unable to tolerate a complete hemodialysis session are at risk for drug accumulation and adverse events.*


The side effect risks of gabapentinoids were likely underrecognized in initial studies supporting their use in enhanced recovery protocols. Because of binding to the alpha-2-delta subunit of N-type calcium channels, which are widely expressed in the hippocampus and cerebellum, gabapentinoids cause dizziness, balance disorders, ataxia, visual disturbances, sedation, somnolence, and cognitive impairment. Gabapentinoids in combination with other central nervous system (CNS) depressants, such as opioids, have an increased risk of noninvasive ventilation and naloxone use in the perioperative setting [77]. In the general population, concomitant use of gabapentinoids and opioids increases the risks of opioid-related death and hospitalization in patients on dialysis [78]. In 2019, the U.S. Food and Drug Administration (FDA) required a warning on the labeling of gabapentinoids concerning the risk of respiratory depression, especially when combined with other CNS depressants or in patients with respiratory risk factors, though this was largely based on observational data [79]. A recent meta-analysis of randomized controlled trials of perioperative gabapentinoid use did not find any significant increase in respiratory depression, though visual disturbances and dizziness were significantly increased with gabapentinoids [80]. The FDA is also requiring new clinical trials of gabapentinoids to assess respiratory depression, particularly in combination with opioids, as additional data from controlled trials are needed [77,81].


**KEY CONSIDERATION: **
*Gabapentinoids are narcotics with dose-dependent adverse effects, which may include respiratory depression, especially with concomitant CNS depressants or in high-risk patients.*


We therefore recommend, when considering a gabapentinoid as part of a multimodal analgesic regimen, careful assessment of patient risk factors for respiratory depression and cumulative exposure to CNS depressants. A “low and slow” approach is also essential to safe and successful use, as adverse event risks are dose-dependent and modified by tolerance. We recommend initiating gabapentin at no more than 300–600 mg per day in gabapentinoid-naïve patients, given in divided doses (i.e., 100 mg three times daily, or 100 mg-100 mg-300 mg regimen), with further downward adjustment in patients of advanced age or renal impairment. Hold parameters for the aforementioned adverse events should also be incorporated into medication orders as appropriate.

## 6. Clinical Pearl #5: Low-Dose Ketamine Is a Powerful Analgesic even in Opioid-Tolerant Patients and Is Generally Well-Tolerated with Appropriate Institutional Protocols

Ketamine is an NMDA antagonist with a host of other pharmacologic properties, resulting in a wide range of effects across its dose–response curve. Ketamine’s complex pharmacology has led to its exploration in a wide variety of therapeutic and psychotropic uses across multiple decades, though its misuse or suboptimal prescribing can lead to significant adverse events. While it has traditionally been used in higher dose ranges (i.e., 2–4 mg/kg) for general anesthesia or tranquilization in acute agitation, ketamine exerts a powerful analgesic effect at low doses (i.e., 0.1–0.3 mg/kg) without conferring respiratory depression, making it an attractive pain management modality [82,83,84,85]. It may be especially advantageous in patients with opioid tolerance, those undergoing highly painful major surgical procedures, or those experiencing a pain-sedation mismatch or other opioid-related adverse events [82,84,86,87,88,89].


**KEY CONSIDERATION: **
*Among available nonopioid analgesics, ketamine has the strongest evidence supporting improved pain control and opioid-sparing properties in patients with preexisting opioid tolerance. It may also interrupt the pathological processes of central sensitization and opioid-induced hyperalgesia, conferring benefit in severe or difficult-to-treat pain.*


Low-dose ketamine, also referred to as analgesic-dose or subdissociative ketamine, has been found to be safe and effective for acute pain in inpatient populations. Randomized controlled trials in the emergency department and postoperative wards have suggested analgesic effectiveness and tolerability comparable to morphine, in addition to opioid-sparing effects [90,91,92]. Published institutional experiences have suggested that analgesic ketamine protocols can be safely implemented on inpatient units, though appropriate caution should be taken to limit adverse events such as hallucinations, dizziness, and sedation [93,94]. Adverse events with ketamine are dose- and administration rate-related, and patients of advanced age are more sensitive to them [82,95,96]. Doses above 0.15 mg/kg may not confer additional analgesic benefit and administration should be by slow intravenous push, intermittent infusion, or continuous infusions [12,84,97,98]. Although adverse cardiovascular or psychotropic effects are not usually problematic with low doses and gentle infusion rates, ketamine may still be unwise in patients with active, severe cardiovascular disease or psychiatric conditions [82,84,86].


**KEY CONSIDERATION: **
*Collaborative, evidence-based institutional protocols and prescriber/staff education are essential to the safe use of analgesic-dose ketamine in hospitalized patients. With appropriate provider knowledge (and/or availability of specialist consultation) and standardized procedures, low-dose ketamine can be safely employed for acute pain management by general practitioners on general medical/surgical inpatient units.*


We therefore recommend that hospital providers familiarize themselves with available guidelines for using low-dose ketamine for acute pain [84], and consider incorporating this modality into multimodal regimens for patients with severe/uncontrolled pain, those with preexisting opioid tolerance, or those experiencing unacceptable opioid-related adverse events. Patients unable to receive low-dose ketamine could be offered magnesium as an alternative NMDA antagonist that may be more familiar and available to hospital prescribers [25,88,99,100].

## 7. Clinical Pearl #6: Empiric Opioid Regimens should include Consideration of Optimal Agent Selection, Dosing, Route of Administration, and Supportive Therapies

When prescribing opioids for acute pain, clinicians will need to consider several factors including the pain type, its severity and location, the speed of analgesic effect necessary for the situation, the analgesic duration of action, ease of use, resources available in the institution, patient preferences/adherence, and cost. The clinician will need to assess not only for the preferred opioid agent, but which route of administration would be the most suitable, what would be the best dosage for that route, and if there are unique limitations and/or contraindications for that patient [101,102].

### 7.1. Opioid Agent Selection

When selecting an empiric opioid regimen for patients without prior exposure to opioids, there are “generally preferred” agents that are more likely to be safe and successful in practice, as described in the framework offered in Figure 2 below. In the absence of widely-available genetic testing, it is rational to choose agents with less genetic variability in response, decreased reliance on end organ function for safety, fewer drug–drug interactions and less histamine release [101,103,104,105,106,107,108,109,110,111,112]. Agent selection should always be tailored in patients with prior opioid exposure history and monitored for adjustment, as discussed in subsequent sections.

### 7.2. Empiric Opioid Dosing

Empiric opioid dosing for acute pain should be stratified based on the patient’s degree of pain or expected pain, preexisting opioid tolerance, and risk for opioid-related adverse events (ORAEs). Opioid tolerance vs. naïvety has historically been considered a binary classification, often demarcated at opioid exposure of ≥60 mg of oral morphine equivalent dose per day for at least 7 days [113,114]. Our collective understanding has since evolved to better appreciate that patients with lower degrees of opioid exposure do develop various degrees of tolerance, and often quickly after opioid exposure. Additionally, other factors can augment a particular patient’s response to opioids and their risk for ORAEs. We therefore recommend a more nuanced approach to assessing prior opioid exposure when managing acute pain, as has been proposed in recent guidelines (Figure 3) [86].

We therefore recommend institutions construct acute pain order sets with various stratified dosing regimens available alongside evidence-based decision support to guide providers in making this determination. The reader is also urged to recognize that the opioid equivalent doses described above are not intended for direct patient care applications such as when changing between opioid agents. The reader is referred to other evidence-based resources when navigating the complexity of converting existing opioid regimens to alternative opioids [11].

### 7.3. Opioid Routes of Administration

The intravenous (IV) route of administration is the fastest route to provide analgesia. Intermittent IV bolus doses as needed (“prn”) are suggested for titration of opioids for severe acute pain since this option provides rapid onset without the uncertainty of medication absorption by other routes. However, the risks and costs of IV agents are higher than their oral counterparts, so IV analgesic regimens should generally be converted to enteral ones as soon as appropriate [102]. This can often happen in the immediate postoperative period for most surgeries in modern enhanced recovery models [12,26]. If the IV route is needed, consideration should be given to the use of patient-controlled analgesia, if institutional processes are in place to support their safe use [20].

Oral administration of analgesic agents has been the gold standard, being simple, noninvasive, demonstrating good efficacy with high patient acceptability, and similar efficacy as the intravenous route [102]. The sublingual (under the tongue) and buccal (placed in the cheek of the mouth) routes of administration can be advantageous if the oral or parenteral routes are not desired or feasible. Highly concentrated opioid solutions are available for sublingual (SL) use and have the advantage of being almost as fast with onset to analgesia as the intravenous route, possibly with lower incidence of opioid-induced respiratory depression [12,115]. Buccal transmucosally absorbed opioids also have a rapid onset of action due to the substantial blood supply to the administration area. The rest of the medication not absorbed sublingually or transmucosally is swallowed and enterally absorbed, where it is subjected to the first pass effect [11]. This amount that would be swallowed is scant and does not contraindicate “nothing by mouth” status, and sublingual opioids can even be employed in intubated patients.


**KEY CONSIDERATION: **
*The route of opioid administration should be tailored to patient status and degree of need for rapid onset, with the enteral route being the safest and easiest when possible. The sublingual route of administration also offers a fast onset of action.*


The subcutaneous route of administration can be chosen if oral, SL, or IV routes are not available or optimal. Examples include combinations of cases of limited venous access, intractable nausea, emesis, dysphagia or bowel obstruction, or if mental status precludes safe oral and SL administration. Intermittent subcutaneous administration can be employed for as-needed opioids, or continuous subcutaneous administration can be pursued in more select cases [116]. Care sites that support this option will have nursing and pharmacy policies regarding insertion of the butterfly needle, care of the site, medications approved for use, priming of the tubing with medication, and incompatibilities of medications that should not be given in the same subcutaneous site. Intramuscular (“IM”) analgesics should not be used for any acute pain management due to wide fluctuations in absorption from the site of injection, a 30–60 min lag time to peak analgesic effects, the potential for nerve injury with multiple dosing, and the availability of multiple preferably routes of administration [11].


**KEY CONSIDERATION: **
*The subcutaneous route of administration for opioids is a valued option in unique situations; the intramuscular route of administration should be avoided.*


### 7.4. Supportive Therapies to Be Co-Prescribed with Opioids

When opioids are being utilized, even if only on an “as needed” basis, a prophylactic bowel regimen should be maintained to minimize opioid-induced constipation [117]. Otherwise, opioid binding occurs in the kappa- and mu-receptors in the enteric nervous system and constipation, nausea, and vomiting are expected [118,119,120,121]. There is limited evidence on which to base the selection of the most appropriate prophylactic bowel regimen. It is recommended that a stimulant laxative, with or without a stool softener or osmotic laxative, with adequate fluid intake, be initiated with the introduction of opioids [120,121]. Bowel regimens are of heightened importance in the postoperative period and in patients otherwise vulnerable to constipation. If the patient has to remain in strict “nothing by mouth” status, rectal administration of an as-needed laxative (e.g., bisacodyl) can be administered. A stool softener alone (i.e., docusate monotherapy) is ineffective and should not be used without a scheduled laxative.


**KEY CONSIDERATION: **
*Always begin a scheduled stimulant bowel regimen when initiating opioid therapy; as-needed laxatives or docusate alone are not sufficient to prevent opioid-induced bowel dysfunction.*


Opioid exposure, especially in the postoperative period, can also elicit the known ORAEs of nausea and vomiting. This can delay discharge from the hospital or surgical center, delay the return to normal activities of daily living after discharge home, and increase costs of care [16]. We therefore recommended co-prescribing an appropriate as-needed antiemetic whenever opioids are employed for acute pain.

## 8. Clinical Pearl #7: Pain Regimens Should Be Evaluated and Adjusted at Least Daily through Multidimensional Pain Assessments to Optimize Efficacy and Safety Endpoints

Several validated pain assessment tools are available for use when evaluating pain regimens (Table 2) [122,123,124]. An often overlooked, yet important part of pain assessment is effective patient communication. Health literacy and the patient’s primary language must be considered. A language service line utilizing an interpreter trained in the use of medical language should be utilized if the patient and provider do not speak the same language. Patient family members and friends should not be used to interpret and providers should not assume the patient understands requests during bedside assessment.

However, analgesic regimens should not be adjusted based on pain scores alone. A consistent systematic approach that assesses pain location, quality, severity, and response to therapy will help ensure a complete picture of the patient’s pain experience (Figure 4). Patient functionality, such as ability to participate in physical therapy or occupational therapy, and the presence or absence of adverse effects should be taken into consideration in conjunction with pain scores. Additionally, patient goals and expectations should be routinely discussed. An important question to ask is, “what does the patient hope to be able to do that pain is preventing them from doing?” Total absence of pain may not be a realistic goal when balancing efficacy and safety. It is important to frame treatment success from a realistic functionality viewpoint rather than presence or absence of pain.

When assessing regimen efficacy, a working knowledge of expected time to peak drug effect and duration of drug effect can be helpful in making decisions about medication adjustments. Assessing the patient’s pain control, and adverse effects, at the expected time to peak can help determine if the dose chosen is adequate. If the patient’s pain is not well controlled at the expected time to peak, we should not expect it to be controlled for the remaining expected duration of effect. Expected time to peak and drug duration is dependent upon route of drug administration and formulation (Table 3).

A framework for guiding patient-specific opioid adjustments is offered in Figure 5. If the patient’s pain is unchanged at expected time to peak during a severe acute painful episode in a monitored hospitalized setting, consider administering an additional dose that is 50–100% higher than the initial dose. This may be repeated for two to three cycles until pain is tolerable. If the patient’s pain is decreased at the expected time to peak but still not adequately controlled, consider administering an additional dose at the same dose as the initial dose [126].


**KEY CONSIDERATION: **
*Assessing for drug efficacy and adverse effects at the expected time to peak after drug administration is the best way to determine whether the selected dose is right for the patient. Carefully assess the nature of breakthrough pain in order to determine the best approach to regimen adjustments.*


## 9. Clinical Pearl #8: Patients with Opioid Tolerance, Chronic Pain, and/or Opioid Use Disorder Require Higher Opioid Doses and More Supportive Therapies to Achieve Positive Outcomes

Before discussing treatment approaches to the opioid tolerant patient, it is important to distinguish between some key terms: tolerance, dependence, and opioid use disorder (OUD). Opioid tolerance and dependence are natural physiologic processes associated with chronic opioid use. Tolerance occurs when repeated exposure to the same dose of opioid results in diminishing effects. Opioid dependence is evidenced by withdrawal syndrome upon abrupt discontinuation, rapid dose de-escalation, or administration of an opioid antagonist [127]. Opioid use disorder is a relapsing and remitting neurobiologic disease involving multiple factors, including genetic, psychological, and physical components [127,128]. These terms are not interchangeable, and inappropriate use may lead to misplaced patient labels and assumptions. Regardless of substance use history, pain assessment and treatment should be provided for all patients without judgment. Additionally, the use of person-first language and optimal terminology among healthcare providers is important to successful patient care plans [129].


**KEY CONSIDERATION: **
*A relationship of mutual trust and respect is vital to manage the patient’s pain. Chronic pain comes with complex psychological impacts that can inform the patient’s physical pain experience and can affect treatment response.*


As with opioid-naïve patients, optimization of non-opioid analgesics should still be the goal for opioid-tolerant patients. It is important to take a thorough inventory of the patient’s baseline opioid regimen, as well as adjuvants being used. Assess whether this acute pain episode is an exacerbation of the patient’s chronic pain, or an acute injury or surgery unrelated to the chronic source of pain. Gauge whether this pain episode is expected to change the patient’s pain baseline or if chronic pain is likely to continue despite resolution of the acute episode. When determining a pain regimen, total daily opioid requirements should be taken into account, both around-the-clock and as-needed administrations. Whenever possible, the patient’s home regimen (or an equianalgesic equivalent) should be continued. This will help avoid withdrawal symptoms through the iatrogenic creation of a relative opioid deficit and maintain the patient’s baseline level of pain control [130]. However, the patient’s chronic pain regimen should not be expected to be sufficient to manage an acute pain episode. Patients who have been chronically exposed to opioids often have unpredictable responses to opioids and require higher opioid doses to effectively manage pain [130]. More frequent assessment for dosing adjustments should be made. Pain scores are likely to be higher, and acute pain episodes should be expected to resolve more slowly in patients who are chronically exposed to opioids compared to those who are opioid-naïve [130]. Comparison of the patient’s pain scores prior to the acute episode may be beneficial in determining medication efficacy.

Figure 6 offers an example of how one might approach an acute on chronic pain episode in the opioid-tolerant patient. Dose adjustments may be more conservative or aggressive based on patient-specific factors. Factors to consider should include severity of acute pain episode, patient comorbidities, potential organ dysfunction, and other acute medication changes. Close and frequent monitoring should be the rule whenever making medication changes.

## 10. Clinical Pearl #9: When Used for Acute Pain, Patient-Specific Plans for Opioid Tapering and Harm Reduction Should Be Developed and Supported across the Care Continuum

Opioid therapies started for acute pain should arc with the course of acute pain, meaning prescribers should initiate opioids at a thoughtful empiric regimen, titrate to effect and tolerability during acute painful episodes, and then taper the regimen as acute pain resolves. Nonopioid therapies should be maintained throughout this process to optimize analgesia and facilitate opioid weaning [131,132,133]. Patient education prior to hospital discharge is vital for patients being discharged on opioids for acute pain, including how to manage pain at home, expectation-setting for pain resolution and opioid use in the post-discharge timeframe, and opioid safety including safe disposal of unused pills [131,134,135]. Many freely accessible patient education resources are available to providers discharging patients on opioid therapies [136,137].

**KEY CONSIDERATION: ***Opioid tapering goals and regimens must be patient-specific and prospectively, collaboratively discussed with the patient and other care team members in order to be safe and successful. Pre-discharge patient counseling on the pain management plan and opioid safety are essential*.

When acute painful episodes can be anticipated, such as those after a scheduled surgery, prospective patient education should be provided including the anticipated type, severity, and duration of pain, in addition to how pain will be managed and the typical extent of opioid exposure [20]. Evidence-based, procedure-specific guidance is available to inform postoperative opioid prescribing for previously opioid-naïve patients [138,139,140,141,142]. An alarming rate of previously opioid-naïve patients experience persistent opioid use long after anticipated surgical pain has resolved, underscoring the need for intentional pain and opioid management in the postoperative period [5,9,143,144,145,146]. While the development of an opioid use disorder (OUD) is risked with any opioid exposure, this risk appears relatively uncommon even among patients on long-term opioid therapies for chronic pain [147]. There is likely a key association between severe uncontrolled pain and opioid misuse, and current guidelines for opioid-naïve patients advise decisions to maintain postoperative opioid therapy should be made in the context of optimizing pain management and preventing persistent postsurgical pain [148,149]. Evidence-based strategies for tapering opioids, preventing persistent postsurgical pain and opioid use, or mitigating the risk of OUD are currently lacking, but recommended best practices include preoperative pain education and optimization of risk factors, intentional opioid prescribing tailored to anticipated duration of significant pain and patient-specific requirements, and focused efforts to optimize nonopioid multimodal analgesia after hospital discharge [9,20,149].

Opioid tapering strategies after acute painful episodes should be very different for patients without prior opioid exposure than for patients with preexisting opioid tolerance and/or those on chronic pain therapies or MOUD (Table 4). In opioid-tolerant patients, tapers will need to be more gradual and more supportive therapies and care coordination may be necessary to ensure the best possible patient outcomes [131,150]. It is imperative to avoid the iatrogenic reintroduction of severe uncontrolled pain throughout this process to avoid self-directed discharge and ensuing increases in morbidity and preventable healthcare utilization [148]. Chronic pain and/or MOUD therapies should be maintained throughout opioid adjustments after acute painful episodes in concert with the appropriate co-providers. The ultimate opioid regimen targets will vary based on the patient-specific factors, and are best directed by the patient’s outpatient prescribers and care teams.

Alongside opioid tapering plans, hospital providers should consider appropriate harm reduction strategies for patients being discharged on opioid therapies for acute pain. Concomitant narcotics, including benzodiazepines, gabapentinoids, and sedative-hypnotics should be assessed [151,152,153]. Opioid education and take-home naloxone distribution has been found to be feasible and effective in decreasing opioid-related emergency department visits and opioid overdose death rates in a variety of settings [154,155,156,157]. Calls for broadening access to such interventions have been increasing in the wake of the opioid epidemic, and many open access resources exist to support hospital providers in providing opioid education and take-home naloxone to their patients. Co-prescribing take-home naloxone has been recommended for patients with concomitant narcotics (e.g., benzodiazepines), those requiring higher doses of opioids, those at increased risk for opioid-induced respiratory depression (e.g., chronic respiratory conditions), those with a personal history of substance use disorder (s), and anyone at high risk for experience or responding to an opioid overdose [136,158,159,160,161,162].


**KEY CONSIDERATION: **
*Co-prescribing take-home naloxone is evidenced-based harm reduction and should be widely considered by hospital providers prescribing opioids at discharge, alongside appropriate educational and continuity of care efforts.*


We therefore recommend hospital providers discharging patients on opioid therapies for acute pain consider patient-specific goals and therapy plans for the post-discharge phases of care, including the intended use and duration of opioid and nonopioid pain therapies, patient and caregiver education, co-prescription of take-home naloxone, and pain management-related follow-up as appropriate.

## 11. Clinical Pearl #10: Methadone and Buprenorphine Should Almost Always Be Continued throughout Acute Pain Episodes, but Naltrexone Must Be Stopped

The percentage of hospitalized patients on chronic medication for opioid use disorder (MOUD) is increasing, and hospital providers must therefore understand the fundamental concepts relating to managing pain in this high-risk population [163]. There are currently three medications widely utilized for the treatment of OUD: methadone, buprenorphine (single product or in combination with naloxone), and naltrexone (oral or depot intramuscular formulation). It should also be noted that methadone and buprenorphine can also be employed in chronic pain management in the absence of OUD, and naltrexone-containing products are indicated for a variety of non-OUD indications. Each medication has a distinctive mechanism of action and poses unique challenges for acute pain management [164].

Buprenorphine is a partial mu-opioid receptor agonist and kappa-opioid receptor antagonist FDA-approved in 2002 to treat OUD. It is effective for management of cravings due to its high mu receptor affinity and slow release [165] Buprenorphine is available in single-ingredient formulations or in combination with naloxone in a 4:1 ratio. It is important to understand that naloxone is co-formulated with buprenorphine *not* to manage OUD, but rather to reduce the risk of diversion of buprenorphine-containing products, i.e., the crushing and injecting of the dosage form to achieve opioid-induced euphoria. Naloxone undergoes tremendous first-pass hepatic metabolism (approximately 97–98%), resulting in very low oral bioavailability when taken as intended, and can even be co-administered with other opioids in the treatment of opioid induced constipation [166] A common misconception in the management of patients prescribed the combination buprenorphine/naloxone product is the need to change to the singular buprenorphine product. In actuality, the pharmacokinetics of naloxone allow for the continuation of the combination product, which can be helpful in maintaining a patient’s home OUD regimen across transitions of care.

**KEY CONSIDERATION**: *Buprenorphine/naloxone combination products DO NOT have to be changed to buprenorphine-only products during acute painful episodes due to the low bioavailability of naloxone.*

Methadone is a full mu-opioid receptor agonist and NMDA antagonist approved for the management of OUD in the clinic setting. Doses for management of OUD vary from 60–120 mg per day with some patients requiring a higher total daily dose (TDD) to manage cravings and suppress withdrawal symptoms [164]. Barring temporary interruption for true contraindications (e.g., acute respiratory failure without a secured airway), methadone should be continued at hospital admission after verification with the Opioid Treatment Program (OTP).

While it may seem counterintuitive, there is ample published evidence and experience to support the continuation of methadone and buprenorphine regimens during acute painful episodes, including through surgical encounters. Continuation of MOUD improves pain control, reduces the risk of relapse, and has been shown to decrease as-needed opioid requirements in this patient population [167]. A small cohort study of 131 patients, with 74 patients who were continued on buprenorphine and 57 patients with buprenorphine discontinued, demonstrated a lower median oral morphine equivalent (OME) for those patients whose buprenorphine was continued (11 mg vs. 103 mg). The patients who were continued on buprenorphine also had a lower maximum 24 h opioid utilization (60 mg vs. 240 mg) while the median pain levels were similar in each group [168]. Another small cohort study of 55 patients perioperatively, demonstrated increased OME dispensed in the outpatient setting for 60 days following surgery (229 mg in the continuation group vs. 521 mg in the discontinuation group) and average pain scores in the buprenorphine continuation group were lower than the discontinuation group (2.9 vs. 7.6) [169]. A final cohort study of 51 patients compared opioid requirements in the first 24 h after surgery among patients receiving buprenorphine (*n* = 22) and methadone (*n* = 29). The study found no differences in pain scores between the two groups nor no significant differences in patient-controlled analgesia requirements between the two groups or with those patients who did not receive methadone on the day of surgery. However, as found with the previous studies, patients who were not given buprenorphine the day after surgery used significantly more patient-controlled analgesia compared to those who received their dose [170]. Furthermore, current clinical practice guidelines recommend the continuation of methadone and buprenorphine throughout the perioperative period, recognizing the detrimental effects of interruption upon postoperative pain control, opioid use, and recovery [171,172].

**KEY CONSIDERATION**: *Methadone and buprenorphine should be continued during acute painful episodes, including surgical encounters, alongside opioid-tolerant doses of as-needed pain medications within multimodal analgesic regimens.*

Both methadone and buprenorphine have high affinity at the mu opioid receptors, providing suppression of opioid withdrawal symptoms and cravings for 24–48 h, making them ideal therapies for OUD [173]. The duration of analgesia for methadone and buprenorphine is significantly shorter, however, at 4–8 h. Because of the limited analgesic duration of these medications, splitting the home dose of methadone or buprenorphine across multiple doses throughout the day can help manage acute pain and should be considered as part of the multimodal analgesic regimen for patients with OUD experiencing acute pain.


**KEY CONSIDERATION: **
*The duration of analgesia from buprenorphine and methadone is shorter than their duration of reducing cravings and withdrawal symptoms- splitting the home dose throughout the day should be considered as part of the multimodal analgesic regimen during periods of acute pain. The patient’s usual dosing regimen can then be resumed once acute pain has subsided and/or at hospital discharge.*


Naltrexone is a competitive mu-opioid receptor antagonist that blocks the effects of endogenous and exogenous opioids. It is available for the treatment of OUD in a depot intramuscular (IM) injection dosed once monthly [164]. It is also important to note that naltrexone is available as an oral tablet used for alcohol use disorder and in combination with bupropion for weight loss. After an oral dose of 50 mg of naltrexone, 95% of cerebral mu-opioid receptors are occupied, and after 7 days of continuous oral use, the half-life of naltrexone is about 10 h [174]. This creates a significant challenge with acute pain crises as the patient will experience little, if any, effect from opioid therapy. Nonopioids, such as ketamine and regional anesthetic strategies, should be the backbone of analgesic regimens in such patients if unexpected severe pain is incurred and/or if emergency surgery is required. If naltrexone is still active and opioids are required, extremely high doses of opioids can be attempted to out-compete the naltrexone, however, the patient will require close monitoring to ensure oversedation does not occur, especially as the naltrexone begins to wear off [175,176].

The primary strategy to manage pain on those patients who are taking naltrexone is to stop naltrexone prior to the planned procedure. For oral naltrexone, therapy should be stopped 72 h prior to the procedure to allow for the medication to be metabolized and to increase the number of available opioid receptors. For patients receiving IM naltrexone, the dose should be held for at least 30 days prior to elective surgery [176]. It is important to remember that as naltrexone is metabolized, patients will quickly lose their tolerance to opioids and should be monitored closely when opioids are reintroduced or adjusted. This exaggerated response is due to upregulation of opioid receptors when naltrexone therapy is being held, which increases sensitivity to opioids [164] Coordination with the patient’s naltrexone provider is key after elective surgery to ensure the patient’s naltrexone therapy is restarted appropriately after the painful event. Usually, a patient will need to be 2–3 days free of opioids and require a test dose of naltrexone prior to resuming therapy.

**KEY CONSIDERATION**: *Naltrexone-containing products must be stopped prior to any scheduled elective procedures to ensure adequate pain management for patients. Patients who have naltrexone interrupted will rapidly lose their tolerance to opioids and should be monitored closely when opioid therapies are needed.*

*These management recommendations for MOUD during acute painful episodes are summarized in*Table 5*. Additionally,* patients with OUD who are in acute pain should be afforded safe and effective multimodal analgesia by hospital providers, including appropriate as-needed opioids dosed for tolerance as discussed previously. Additionally, coordination of care with the patient’s Addiction Medicine team should be standard of care, to the extent possible. The patient should also always be prospectively included when discussing pain management strategies, especially those patients with OUD. For example, the team may, unknowingly, prescribe a specific opioid which may trigger a relapse and is not preferred by the patient based on their previous drug of choice. Holistic care is important for management of pain crises as it allows for better understanding of the patient, their risk factors, and what you, the clinician, are able to do to help meet pain goals.

## 12. Conclusions

Acute pain management in hospitalized patients can challenge inpatient healthcare providers, but increased familiarity with core concepts and resources can support a consistent and effective approach to daily practice. We hope the pearls and tools offered in this piece help diverse hospital-based prescribers and staff in providing better pain management and improving patient outcomes.

## Figures and Tables

**Figure 1 healthcare-11-00034-f001:**
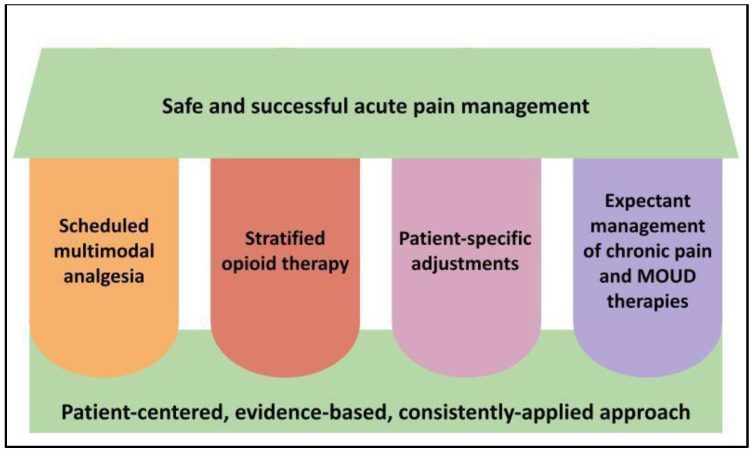
**Key pillars of acute pain management and opioid stewardship in hospitalized patients**. Legend: The four vertical pillars represent our core recommended strategies to apply to acute pain management which, when used in the context of a foundational approach as represented at the bottom of the figure, should support the desired outcome at the top of the figure. MOUD = medications for opioid use disorder.

**Figure 2 healthcare-11-00034-f002:**
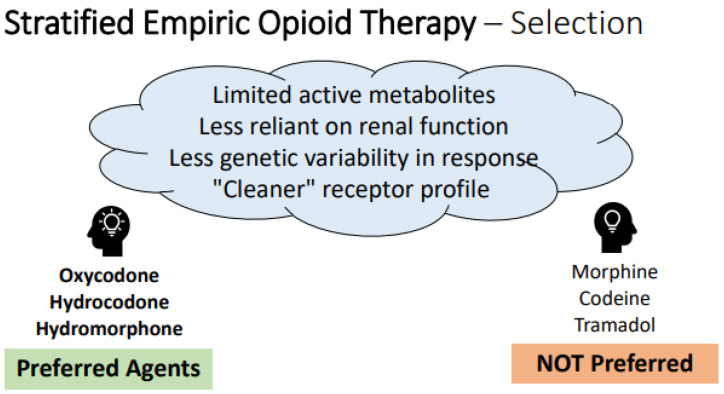
**Framework and recommendations for empiric opioid selection.** Legend: Key determinants of “ideal” empiric opioids are represented in the thought cloud at top, with specific preferred agents listed at left and non-preferred agents listed at right, based on these qualities. Note- this framework only applies to empiric decision-making in the setting of no prior opioid exposure; these considerations must be melded with patient-specific information and exposure history where applicable.

**Figure 3 healthcare-11-00034-f003:**
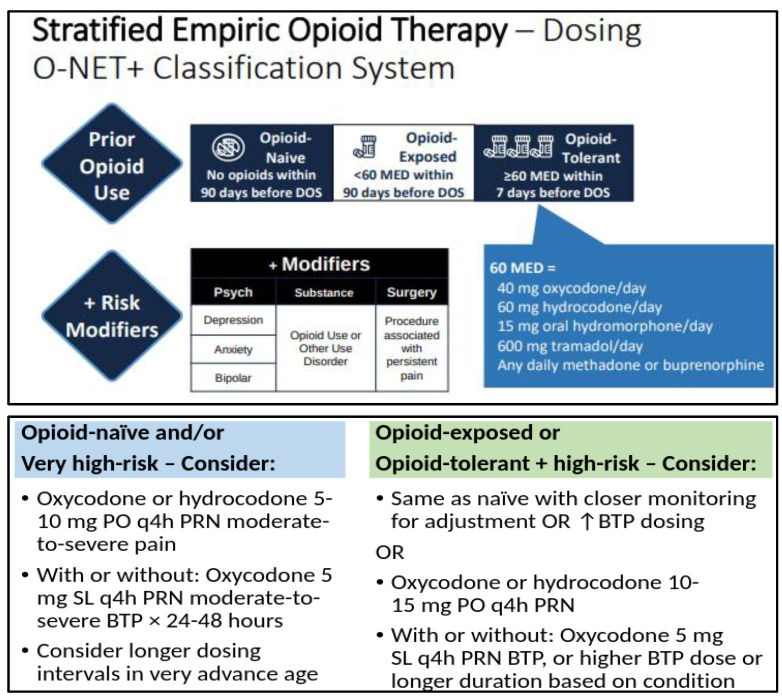
**Framework for empiric opioid dosing considerations, based on an adaptation of the “Opioid-naïve, -Exposed, or -Tolerant plus Modifiers” (O-NET+) stratification system for risk of postoperative opioid related adverse events in patients on preoperative opioid therapy** [86]. Legend: Top pain describes the O-NET+ classification system based on prior opioid use and risk modifiers, and bottom pain describes a recommended risk-stratified approach to initial opioid dosing based on this classification system (see text and reference for further information). BTP = breakthrough pain, DOS = day of surgery, MED = morphine equivalent dose, PO = by mouth, PRN = as needed, SL = sublingual.

**Figure 4 healthcare-11-00034-f004:**
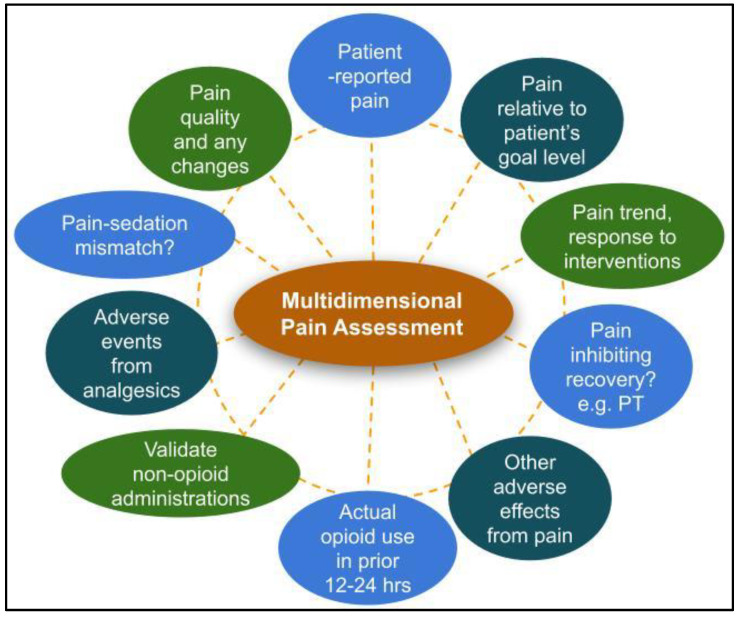
Recommended components of an effective multidimensional pain assessment. Legend: PT = physical therapy.

**Figure 5 healthcare-11-00034-f005:**
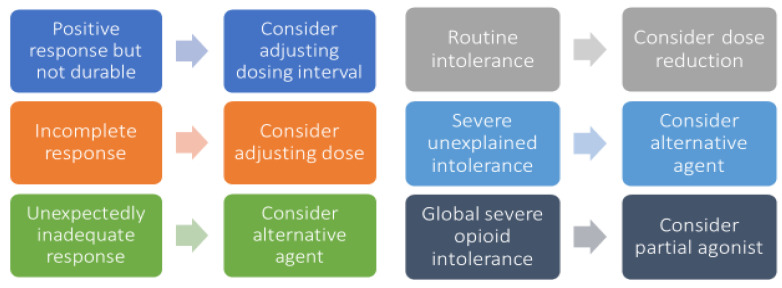
**Framework for patient-specific opioid regimen adjustments.** Legend: each matching pair of boxes represents a potential scenario for opioid regimen optimization, with a specific issue listed in the left box and the recommended adjustment in the associated right box.

**Figure 6 healthcare-11-00034-f006:**
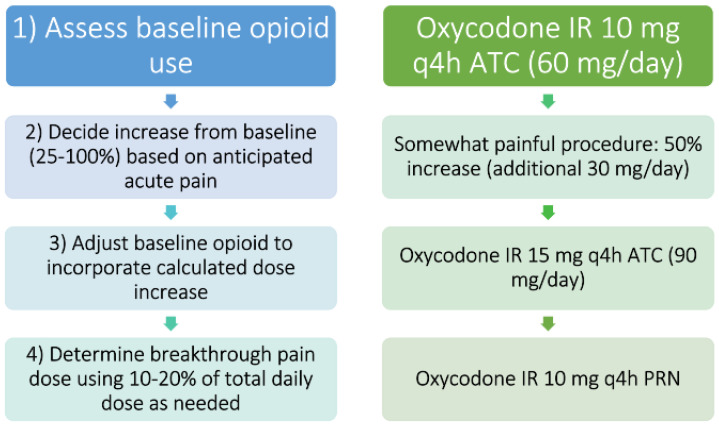
**One process and a worked example for constructing an acute pain regimen in a patient with preexisting around-the-clock opioid therapy and tolerance.** Legend: At left is a step-wise description of the recommended process, with a worked example paralleled at right. Note- this is one example and will not be ideal for all patients. ATC= around the clock, h = hours, IR = immediate release, q = every, PRN = as needed.

**Table 1 healthcare-11-00034-t001:** Concerns and evidence relating to NSAIDs for acute pain management in the inpatient setting.

Concern	Evidence	Recommendation
Bleeding/antiplatelet effects	Bleeding times and perioperative bleeding events are not significantly affected by NSAIDs at usual doses; GI complications from NSAID-induced prostaglandin inhibition are not increased by short-term use (<7 days); These risks may be further mitigated by using a COX-2 selective agent since antiplatelet effects are mediated by COX-1 inhibition	Do not withhold NSAIDs in acute pain due to bleeding concerns as long as usual analgesic doses and short-term durations are employed; selective COX-2 inhibitors or concomitant gastroprotective agents may be considered in patients at high GI bleed risk
Wound healing issues or orthopedic/spinal nonunion after fracture or fusion surgery	Older data from animal and limited retrospective studies suggested these concerns, however more recent and higher quality prospective studies have not replicated	NSAIDs, especially COX-2 selective agents, appear efficacious and safe for short-term use in orthopedic and spinal surgery and should be routinely considered based on risks/benefits
Anastomotic leak after GI surgery	Some studies have suggested increased risk of anastomotic leakage with nonselective NSAIDS, but selective COX-2 inhibitors were not associated with this risk in recent meta-analyses	Do not withhold COX-2 selective NSAIDs in GI surgery patients
MACE after cardiac surgery	COX-2 selective inhibitors have been associated with increased rates of MACE after cardiac surgery, likely due to an unfavorable effect on pro-thrombotic pathways	COX-2 selective agents should be avoided in cardiac surgery, however, nonselective NSAIDs have been used safely in cardiac surgery, and COX-2 selective agents have been used safely in patients with cardiac disease undergoing noncardiac surgery
Sulfa allergy	While some NSAIDs contain a sulfur-containing moiety, these are not structurally the same as sulfa antibiotics; patients with sulfa allergies have been found to be no more likely to have allergic reactions to NSAIDs than patients without sulfa allergies	Do not withhold NSAIDs, including celecoxib, in patients with sulfa (sulfonamide antibiotic) allergies
Gastritis/pouchitis in patients s/p bariatric surgery	Patients s/p bariatric surgery should avoid chronic NSAID exposure, however, short-term use is supported by current guidelines as safe and beneficial	Do not withhold short-term NSAIDs in acute pain in patients s/p bariatric surgery; use of a COX-2 selective agent and/or temporary PPI therapy may be considered to decrease GI risk
Kidney injury	NSAIDs inhibit prostaglandin-dependent mechanisms of preserving renal perfusion and GFR in times of decreased renal blood flow, increasing risk for acute and chronic kidney injury in at-risk populations	All NSAIDs and COX-2 inhibitors should generally be avoided in patients with AKI or CKD
Large doses must be used for analgesia	The maximum effective analgesic dose of ketorolac is approximately 10–15 mg and is approximately 400 mg for ibuprofen based on available dose-finding studies, though higher doses may confer additional anti-inflammatory benefit	When using NSAIDs primarily to treat pain, doses should generally not exceed their analgesic ceiling in order to limit adverse effects

Legend: AKI-acute kidney injury, CKD = chronic kidney disease, COX = cyclooxygenase enzyme, GI = gastrointestinal, MACE = major adverse cardiac events, NSAID = non-steroidal anti-inflammatory drug, PPI = proton pump inhibitor, s/p = status post. References: [45,46,47,48,49,50,51,52,53,54,55,56,57,58,59,60,61,62,63,64,65,66,67,68,69,70,71,72].

**Table 2 healthcare-11-00034-t002:** Select Available Pain Scales.

Pain Scale ^1^	Description	Intended Population
Visual Analog Scale	numerical scale rating 1–10	adults who are able to self-report pain
Wong-Baker Faces	scale utilizing facial expressions linked to pain severity	patients age 3 and above
Pain Assessment in Advanced Dementia (PAINAD)	utilizes non-verbal cues to assess pain	patients with dementia, unable to self-report
Behavioral Pain Scale	observational assessment	critically ill, sedated patients
Defense and Veterans Pain Rating Scale (DVPRS)	combination graphic and numerical tool	adults who are able to self-report pain

^1^ This is not an exhaustive list of available validated pain assessment tools. References: [122,123,124,125].

**Table 3 healthcare-11-00034-t003:** Estimated opioid pharmacokinetic parameters pertinent to pain regimen monitoring and adjustment.

Formulation/Route of Administration	Time to Peak1—Assess for Efficacy and Adverse Effects	Additional Considerations
Intravenous	10–15 min	
Subcutaneous	30 min	
Immediate release oral	60 min	
Immediate release sublingual	15–30 min	Assess patient ability to hold medication under the tongue

Expected time to peak based on route of administration in otherwise healthy patients without organ dysfunction. Reference: [11].

**Table 4 healthcare-11-00034-t004:** Recommended considerations for tapering opioid regimens after acute painful episodes.

Regimen Component	Approach for Opioid-Naïve	Approach for Opioid-Tolerant
Goals of opioid tapering	Limit excess exposure to opioids and opioid-related adverse events once pain is improving, limit conversion to persistent opioid use if not otherwise indicated by patient condition, limit quantity of unused opioids	More complex and patient-specific, may entail tapering back to previous chronic pain or MOUD regimen (or reevaluating chronic regimen in concert with applicable prescriber), limiting opioid-related adverse events, avoiding relapse of OUD, limiting long-term adverse events related to chronic opioid exposure
Dose reduction at each step of taper	Consider decreasing daily dose by 20–25%	More gradual reductions may be needed at each step
Frequency of tapering	Every 1–2 days once pain is improving	Less frequent reductions are likely to be needed, consider every 2–7 days once acute pain improving
Total duration of taper	Most patients can successfully taper off opioids within 3–7 days after a major scheduled surgery, assuming multimodal and enhanced recovery techniques are used concurrently	Longer tapers will be needed, may take weeks to months to be successful depending on patient-specific circumstances
Other considerations	Consider reducing dose before lengthening dosing interval to help maintain smoother pain control without large peaks/valleys of analgesic effect	More multimodal therapies, psychosocial support, monitoring, and coordination of care often needed

Legend: MOUD = medication(s) for opioid use disorder, OUD = opioid use disorder. References: [11,22,131,138].

**Table 5 healthcare-11-00034-t005:** Management of MOUD during acute painful episodes.

Medication	Mechanism of Action	Acute Pain Strategies
Buprenorphine	Partial mu-opioid agonist, kappa-opioid antagonist	Continue home regimen; Split home regimen into TID dosing for same TDD
Methadone	Full mu-opioid agonist, NMDA antagonist	Continue home regimen; Split home regimen into TID dosing for same TDD
Naltrexone (IM)	Mu-opioid antagonist	Stop IM dose 30 days prior to painful procedure and until patient has been opioid-free for 3 days afterward; Multimodal therapies to treat painful crisis, consider ketamine and regional anesthesia
Naltrexone (PO)	Mu-opioid antagonist	Stop therapy 72 h prior to painful and until patient has been opioid-free for 3 days afterward; Multimodal therapies to treat painful crisis, consider ketamine and regional anesthesia

Legend: IM = intramuscular (depot formulation), TDD = total daily dose, TID = three times daily. References: [164,171,172,176,177].

## Data Availability

Not applicable.

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
