# Peer review of "Acute Pain Management Pearls: A Focused Review for the Hospital Clinician"

_healthcare, 2022, doi:10.3390/healthcare11010034_

Round 1

Reviewer 1 Report

The authors of the work have clearly drawn the clinical aspect linked to the management of acute pain, a very difficult field in its management and patient care is increasingly complex and expensive in this area.

The aspect of patients developing addictions needs to be broadened

Author Response

The authors of the work have clearly drawn the clinical aspect linked to the management of acute pain, a very difficult field in its management and patient care is increasingly complex and expensive in this area.

Response: Thank you for your review and feedback

The aspect of patients developing addictions needs to be broadened

Response: This is a fair point. We have added text to this discussion in section 10 (Clinical Pearl #9), second paragraph. 

Reviewer 2 Report

This paper is really instructive, since the authors outline acute pain treatment and offer recommendations for professionals in clinical practice. I just have one question. Multiple non-pharmacological treatments were discussed on line 149; could the authors provide guidance as to which alternative is most suitable, e.g., should these practices be utilized concurrently or sequentially with other pharmacological therapies?

Author Response

This paper is really instructive, since the authors outline acute pain treatment and offer recommendations for professionals in clinical practice. I just have one question. Multiple non-pharmacological treatments were discussed on line 149; could the authors provide guidance as to which alternative is most suitable, e.g., should these practices be utilized concurrently or sequentially with other pharmacological therapies?

Response: Thank you for your review and feedback. We have added additional guidance in the final paragraph of section 3 (Clinical Pearl #2) to add clarity here.

Reviewer 3 Report

In this article, the authors described about acute pain managements and presented the ten clinical pearls. I think this review is substantially well-described, and will be helpful for a lot of hospital clinicians.

There are some minor concerns which I want authors to improve before publication.

1. On the whole, authors did not specify the pain focused on the manuscript. It might mainly be a nociceptive pain; if so, authors ought to clearly describe about it. Or, if it included a neuropathic pain and others, some descriptions would be confusing and lead to misunderstand.

2. Some figures are difficult to understand by just themselves because they lack appropriate legends. Some are just lists of phrases or sentences, and don’t have values to present as figures. I think they should be improved to visually understand.

Author Response

In this article, the authors described about acute pain managements and presented the ten clinical pearls. I think this review is substantially well-described, and will be helpful for a lot of hospital clinicians. There are some minor concerns which I want authors to improve before publication.

Response: Thank you for the review and feedback

1. On the whole, authors did not specify the pain focused on the manuscript. It might mainly be a nociceptive pain; if so, authors ought to clearly describe about it. Or, if it included a neuropathic pain and others, some descriptions would be confusing and lead to misunderstand.

Response: This is a fair point. We have elaborated on this and referred the reader to additional resources as appropriate at the end of section 1 to clarify this. 

  1. Some figures are difficult to understand by just themselves because they lack appropriate legends. Some are just lists of phrases or sentences, and don’t have values to present as figures. I think they should be improved to visually understand.

Response: This is a fair point. We have elaborated the Legends of Figures 1, 2, 3, 4, 5, and 6. We have also added borders to the figures and recreated Figure 4 to aid in readability and understanding of the relationships/process represented.